# The Relationship between Dietary Pattern and Bone Mass in School-Age Children

**DOI:** 10.3390/nu14183752

**Published:** 2022-09-11

**Authors:** Xuemei Liao, Shanshan Chen, Mengyang Su, Xuanrui Zhang, Yuanhuan Wei, Shujun Liang, Qinzhi Wei, Zheqing Zhang

**Affiliations:** 1Food Safety and Health Research Center, Guangdong Provincial Key Laboratory of Tropical Disease Research, School of Public Health, Southern Medical University, Guangzhou 510515, China; 2Department of Nutrition and Food Hygiene, Guangdong Provincial Key Laboratory of Tropical Disease Research, School of Public Health, Southern Medical University, Guangzhou 510515, China

**Keywords:** dietary patterns, school-age children, bone mineral content, bone mineral density

## Abstract

Early bone accrual significantly influences adult bone health and osteoporosis incidence. We aimed to investigate the relationship between dietary patterns (DPs), bone mineral content (BMC) and bone mineral density (BMD) in school-age children in China. Children aged six–nine years (*n* = 465) were enrolled in this cross-sectional study. DPs were identified by principal component factor analysis. Total body (TB) and total body less head (TBLH) BMC and BMD were measured using dual-energy X-ray absorptiometry. Five DPs were identified. After adjustment for covariates, multiple linear regression analysis showed that the “fruit-milk-eggs” dietary pattern was positively associated with TB (*β* = 10.480; 95% CI: 2.190, 18.770) and TBLH (*β* = 5.577; 95% CI: 0.214, 10.941) BMC, the “animal organs-refined cereals” pattern was associated with low TB BMC (*β* = −10.305; 95% CI: −18.433, −2.176), TBLH BMC (*β* = −6.346; 95% CI: −11.596, −1.096), TB BMD (*β* = −0.006; 95% CI: −0.011, −0.001) and TBLH BMD (*β* = −0.004; 95% CI: −0.007, −0.001). In conclusion, our study recommends home or school meals should be rich in fruit, milk, eggs with a moderate amount of vegetables, coarse grains and meat to promote bone development for school-age children.

## 1. Introduction

Among the factors associated with osteoporosis, low peak bone mass (PBM) plays an important role. PBM is the maximum bone mass reached at the end of skeletal maturation. It has been reported that a 10% increase in peak bone mineral density (BMD) is associated with a 13-year delay in osteoporosis onset [1]. Bone accumulation occurs during early life, particularly during childhood and puberty, which are crucial periods of bone mass accrual [2]. Thus, childhood bone development influences PBM and the future probability of fractures and osteoporosis. It has been reported that children with reduced bone mass have a greater risk of fractures [3].

Although genetic factors are the primary determinant of PBM, nutrition and lifestyles serve as important modifiers in early life [4]. Therefore, childhood interventions aimed at stimulating bone growth, increasing PBM and preventing fractures are required. As childhood nutrition is considered to be the cornerstone of future bone mass accrual [5], several studies have explored whether specific foods or nutrients can affect bone mass in children. A systematic review indicated that a reduced intake of milk or the consumption of high energy or sugar-sweetened beverages was associated with increased fracture risk in children aged 2–13 years [6]. Similarly, another systematic review reported that the regular consumption of dairy products was associated with increased bone mineral content (BMC) in children [7]. Other studies have examined the relationships between fruit and vegetable [8,9], mineral [10] and vitamin [11] intake and bone status in pediatric populations.

A comprehensive and systematic dietary index would be more appropriate for examining the influence of diet on bone health, given that people consume many different foods and nutrients each day, and the interactions among them can be complex [12]. Moreover, determining the effects of individual nutrients or foods can be difficult, as such effects may be extremely small. Additionally, the identification of dietary patterns based on food intake allows for a better appreciation of the associated nutrients and the incorporation of any findings into future dietary guidelines [13,14].

Childhood represents a critical window for establishing lifelong dietary patterns and healthy habits. Although the importance of childhood dietary patterns for lifelong health is increasingly recognized, only a few studies have been carried out in the USA [15], Spain [16], South Korea [17] and other countries [18] to assess the association between dietary patterns and bone health in pediatric populations, and the results have been inconsistent. These discrepancies can be explained by cultural differences in diet or the use of different study methods to explore such dietary patterns. Little is known about the effects of dietary patterns on bone health among children in China, who are thought to exhibit unique dietary structures [19,20]. For this reason, the aims of this study were to: (1) explore the extant dietary patterns among school-age children, and (2) examine the association between such dietary patterns and bone measurements in school-age children in China.

## 2. Materials and Methods

### 2.1. Participants

Healthy children aged six–nine years were recruited from schools in Guangzhou, China, via invitation letters, advertisements, public WeChat accounts and referrals from December 2015 to March 2017, as reported previously [21]. The exclusion criteria were as follows: (1) history of acute or chronic disease or malnutrition; (2) twins or preterm births; or (3) incomplete data. A total of 465 children (200 females and 265 males) were included in this study. Written consent was obtained from each participant through parents or legal guardians prior to enrollment. The study was approved by the ethics committee of the School of Public Health at Sun Yat-sen University (No. 201549).

### 2.2. Data Collection and Anthropometric Measurements

The parents were interviewed to assess the demographic, socioeconomic and lifestyle characteristics of the participants. Sociodemographic characteristics included age, sex, height, weight and household income. Household income was divided into six levels (<2000, 2000–5000, 5001–8000, 8001–12,000, 12,000–15,000 and >15,000 Chinese yuan (CNY)/month) and unknown. Anthropometric measurements included body weight and height. Children were asked to wear light clothing and no shoes, and all measurements were collected by trained technicians using a standard protocol, as described elsewhere [21]. Weight and height were measured to the nearest 0.1 using a Tanita MC-780A scale (Tanita Corporation, Tokyo, Japan) and a portable fixed stadiometer (model TZG, China), respectively. Body mass index (BMI) was calculated as weight (kg)/height squared (m^2^), while the BMI z-score was obtained according to a World Health Organization (WHO) growth reference for school-aged children and adolescents [22]. Lifestyle characteristics included physical activity and passive smoking. Daily physical activity was estimated using a three-day physical activity questionnaire (including two weekdays and one weekend day) and was calculated by summing the metabolic equivalent scores (MET, kcal/kg/h) for each type of physical activity after multiplying it by its duration per day (h/day) [23].

### 2.3. Assessment of Dietary Intake

The average dietary intake of the participants over the previous year was assessed using a 79-item food frequency questionnaire (FFQ). The reproducibility and validity of the FFQ were described previously [24]. The frequency of consumption (never, daily, weekly, monthly and yearly) was estimated for various foods. Photographs of the food in standard portion sizes were provided to aid parents in estimating the amount typically consumed. The chosen frequency and portion size for each food item was then used to calculate the average daily intake for each item. Total daily energy intake was calculated according to the 2009 China Food Composition Table [25].

### 2.4. Measurement of BMC and BMD 

Whole body scans were captured using dual-energy X-ray absorptiometry (Discovery W; Hologic Inc., Waltham, MA, USA) to obtain the TB BMC, TBLH BMC and BA values, as per the manufacturer’s instructions. BMD was calculated by dividing BMC by BA. Repeatability was assessed by repositioning and reimaging 33 participants on the same day. The coefficients of variation of the duplicated BMC (BMD) measures for TB and TBLH were 1.09% (1.58%) and 1.37% (2.04%), respectively.

### 2.5. Statistical Analysis

Continuous variables are expressed as means and standard deviations (SD) if normally distributed, or as medians with interquartile ranges if not normally distributed. Categorical variables are presented as numbers and percentages. 

Dietary patterns were identified using factor analysis of 21 food groups. First, the Kaiser–Meyer–Olkin (KMO) test and Bartlett’s test of Sphericity were carried out. The KMO value was 0.648, and the significance of Bartlett’s Sphericity was below 0.05, indicating that factor analysis was appropriate to use. Factors were rotated by varimax rotation to maintain independent factors while improving interpretability. The main factors (i.e., dietary patterns) were selected based on the eigenvalue (>1.25), scree plot, factor interpretability and proportion of variance explained by each factor. The dietary patterns were labeled according to high food group loadings (absolute value) for each factor. Each dietary pattern was assigned a factor score for each participant, which was calculated by summing the intake of each food group weighted by their factor loadings. A higher score indicates a higher adherence to the respective dietary pattern. For each dietary pattern, the participants were categorized into tertiles according to the residual adjusted dietary pattern scores. 

Multiple linear regression analyses were then used to estimate the association between BMC/BMD and dietary pattern scores among the participants. Model 1 was adjusted for age, sex and energy intake. Model 2 was additionally adjusted for height, weight, physical activity, household income, parental education, dietary protein intake, dietary calcium intake, dietary vitamin D intake, supplemented intake of calcium (yes or no), supplemented intake of multivitamins (yes or no) and passive smoking (yes or no). Additionally, analysis of covariance (ANCOVA) was used to compare the covariate-adjusted BMC and BMD tertile means by dietary pattern scores. Pair-wise comparisons were performed using the Bonferroni test. The percentage differences in TB and TBLH BMC and BMD between the first and third tertile of the dietary pattern scores were calculated. Linear trend analysis was conducted using a general linear model to investigate association of BMC/BMD with each dietary pattern. All statistical analyses were performed with IBM SPSS Statistics version 20.0 software (SPSS Inc., Chicago, IL, USA). The significance level was defined as *p* < 0.05 and all *p*-values were two-sided.

## 3. Results

### 3.1. Characteristics of Participants

A total of 465 children were included (200 females and 265 males) with a median age of eight years. The descriptive characteristics of the participants are presented in Table 1. Gender differences in dietary patterns were detected, with boys more likely to report “fruit-milk-eggs” and “animal organs-refined cereals” patterns and girls more likely to report a “seafood-mushrooms-nuts” pattern.

### 3.2. Dietary Patterns Derived from Dietary Intake

Five predominant dietary patterns were identified among the 21 food groups included in the FFQ. Factor loading matrices are shown in Table 2. Dietary pattern 1 was characterized by high factor loadings of fresh vegetables, whole grains and red meat, and was denoted as the “vegetables-whole grains-red meat” dietary pattern. Dietary pattern 2 was characterized by high factor loadings of fruit, full-fat milk and eggs, and was termed the “fruit-milk-eggs” pattern. Dietary pattern 3 was characterized by high factor loadings of mollusks and shellfish, mushrooms and nuts, and was labeled the “seafood-mushrooms-nuts” pattern. Dietary pattern 4 was characterized by high factor loadings of freshwater and sea fish, low-fat milk, soup and beverages, and was denoted as the “beverages-fish-low-fat milk” pattern. Dietary pattern 5 was characterized by high factor loadings of refined cereals, preserved vegetables, red meat, and animal organs, and was labeled the “animal organs-refined cereals” pattern. These five patterns explained 39.9% of the variability. All dietary pattern scores were adjusted by energy intake.

### 3.3. Multiple Linear Regression of Dietary Pattern Scores and Bone Mass

The multiple linear regression analysis of the relationship between dietary pattern scores and bone mineral status is shown in Table 3. In Model 1, the “vegetables-whole grains-red meat” and the “fruit-milk-eggs” patterns were positively associated with all bone measurements (*p* < 0.05). In contrast, the “animal organs-refined cereals” pattern was negatively correlated with all bone measurements (*p* < 0.05). In the fully adjusted model, every unit increase in the “fruit-milk-eggs” pattern score was associated with a 10.48 g (95% CI: 2.190, 18.770) increase in TB BMC and a 5.577 g (95% CI: 0.214, 10.941) increase in TBLH BMC. In contrast, every unit increase in the “animal organs-refined cereals” pattern score correlated with a 10.305 g (95% CI: −18.433, −2.176), 6.346 g (95% CI: −11.596, −1.096), 0.006 g/cm^2^ (95% CI: −0.011, −0.001) and 0.004 g/cm^2^ (95% CI: −0.007, −0.001) decrease in TB BMC, TBLH BMC, TB BMD and TBLH BMD, respectively. No other significant associations between dietary patterns and bone mass were identified.

### 3.4. Analysis of Covariance of BMC and BMD by Dietary Pattern Score Tertiles

ANCOVA analyses of the tertile-wise differences in dietary pattern scores and bone indices are listed in Table 4. After fully adjusting for covariates, the percentage differences in the adjusted mean TB BMC and TBLH BMC values between the first and third tertiles of the “fruit-milk-eggs” pattern were 2.18% and 2.08%, respectively (*p*-trend: 0.021–0.033). Children in the second tertile of “beverages-fish-low-fat milk” had a lower TB BMC and TBLH BMC compared to those in the first or third tertile (*p* = 0.009). No significant association with bone mass was observed for any other dietary patterns.

## 4. Discussion

In our cross-sectional study, five predominant dietary patterns were identified among school-age children in China. We observed that children who had a higher intake of fruit, milk and eggs had a higher TB BMC and TBLH BMC. Additionally, the “animal organs-refined cereals” pattern, characterized by a diet rich in refined cereals, preserved vegetables, red meat and animal organs, was inversely associated with TB BMC, TBLH BMC, TB BMD and TBLH BMD. A U-shaped association with TB BMC and TBLH BMC was noted for the “beverages-fish-low-fat milk” pattern. No other significant associations between dietary patterns and bone mass were identified.

Previous studies examining the impact of diets rich in fruit, milk or eggs on skeletal health have reported results consistent with this study [26,27,28,29]. Movassagh et al. reported that a diet rich in fruit and vegetables was positively associated with TB BMC (*β* = 35.2, *p* = 0.025) in adolescence (age 12.7 ± 2 years) and was further associated with young adult TB BMC (*β* = 55.8, *p* = 0.021) and BA-adjusted BMD (*β* = 0.016, *p* = 0.041) in a prospective cohort study (1991–2011) [29]. Fruit contains abundant antioxidants, such as vitamin C, vitamin K, folate and carotenoids. Such antioxidant properties may contribute to increased bone remodeling and repair. Additionally, fruit is rich in other nutrients such as potassium and magnesium, which may also prevent bone loss by increasing urine pH, thereby increasing calcium reabsorption from the renal tubules [30]. A prospective cohort study of 2850 children observed that infants falling in the highest quartiles of dietary patterns characterized by dairy, whole grains and eggs exhibited a 3.98 mg/cm^2^ greater BMD and a 4.96 g greater BMC in childhood compared to those falling in the lowest quartile [26]. A prospective study with two annual follow-up surveys was conducted in 198 Korean females aged 9–11 years and reported that subjects with higher scores for a dietary pattern characterized by fruit, nuts and seeds, dairy products, eggs and grains experienced a greater increase in calcaneus BMC (*p*-trend < 0.01) over a 22-month period [27]. Dairy products are a source of calcium, magnesium, phosphorus, potassium and zinc. Calcium and phosphorus are bone-forming minerals and exist in the bone and teeth as hydroxyapatite [31]. The consumption of calcium-rich foods has been recommended for its positive effect on increasing bone mineralization within a given bone volume, especially during childhood [32,33]. In addition to calcium, other minerals found in dairy products, such as magnesium, phosphorus, potassium and zinc, might exert beneficial effects on bone mineral accumulation [33]. In a cross-sectional study of 294 US children aged 9–13 years, egg intake was positively correlated with radial (cortical) BMC (r = 0.138, *p* = 0.031) [28]. Eggs contain several nutrients, including calcium, phosphorus, potassium, protein, vitamin D, zinc and bioactive compounds (e.g., lutein and zeaxanthin) [28,34]. Zinc can improve bone mass by increasing the serum concentration of the bone formation marker procollagen type 1 amino-terminal propeptide (P1NP) [35]. Various in vitro experiments have shown that lutein and zeaxanthin exert antiinflammatory effects by reducing the levels of inflammatory cytokines such as tumor necrosis factor α (TNF-*α*) and interleukin 1*β* (IL-1*β*) [36,37,38]. Systemic inflammation can lead to excessive bone loss and inhibit osteoblast maturation, leading to decreased bone mineralization [39]. In a recent review, dietary protein consumption above the current Recommended Dietary Allowance (RDA) (0.8 g of protein per kg of body weight) was considered beneficial in bone protection in older people, provided calcium intakes are adequate [40,41]. Protein has a structural function in bone metabolism [42]. Furthermore, the protein found in both dairy and eggs contributes to the bioavailability of other nutrients. For example, protein can stimulate calcium absorption from the gut [43]. Adequate protein intake can thus improve calcium absorption and promote bone mineralization [44]. Moreover, protein can increase the circulating concentration of insulin-like growth factor-1 (IGF-1), which can improve intestinal calcium and phosphorus absorption and increase phosphate reabsorption from the kidney [34]. Furthermore, dietary proteins provide amino acids to join in several signaling pathways of bone maintenance and protection [45].

Another finding of this study was that the “animal organs-refined cereals” dietary pattern was inversely associated with bone health. In line with our findings, McNaughton et al. identified a dietary pattern (refined cereals, soft drinks, fried potatoes, sausages and processed meat, vegetable oils, beer and takeaway foods) that exhibited an inverse association with TB BMC (*β* = −15.4; 95% CI: −27.4, −3.3) in women aged 18–68 years [46]. While the potential mechanisms underlying this association are uncertain, it has been assumed that potassium can probably suppress calcium resorption or/and bone mineral dissolution to reduce urinary calcium loss and improve calcium balance [47]. Moreover, the “animal organs-refined cereals” pattern identified in this study was rich in preserved vegetables, with considerable amounts of sodium and no potassium, leading to an overall reduction in total dietary potassium content [48]. Furthermore, sodium and calcium ions compete within the renal tubule, such that increased sodium intake can lead to elevated urinary calcium excretion, which needs more calcium intake to compensate [49]. The underlying biological mechanism responsible for animal organ consumption being negatively associated with bone mass may relate to the presence of metallic elements or contaminants (such as Cd and Pb) accumulated in the ingested organs. Such compounds could affect osteoblast and osteoclast balance, inhibit the synthesis of osteocalcin (a marker of bone formation) and induce oxidative stress [50,51,52,53]. However, further studies are required to elucidate the specific mechanisms underlying the effects of such dietary patterns on bone mass.

A U-shaped association was observed between the “beverages-fish-low-fat milk” pattern and bone health. This dietary pattern contains both “healthy” and “unhealthy” foods in terms of bone health. As the associated score increases, it can be divided into tertiles. In the first and third tertile, foods with positive effects on bone health predominated, while in the second tertile deleterious foods predominated. Therefore, a U-shaped relationship between this dietary pattern and bone health was seen. 

We found no association between the “vegetable-whole grains-red meat” or the “seafood-mushrooms-nut” pattern and BMC or BMD. Vegetables represent a rich source of beneficial bioactive compounds including nitrate, phytochemical compounds, potassium, manganese and vitamin K1 [54]. Furthermore, vegetables can decrease dietary acid load and reduce urinary calcium excretion, thus inhibiting bone loss. Whole grains are nutrient-rich and are a source of protein, fiber, B vitamins, antioxidants and trace minerals. Seafood is an excellent source of long-chain omega-3 fatty acids and vitamin D_3_, while mushrooms are the best nonanimal food source of vitamin D_2_. Nuts are an excellent source of minerals. Together, these nutrients provide nutritional support to help build strong bones [55]. Vitamin D is supposed to promote bone formation and growth by regulating calcium and phosphate homeostasis and promoting the formation and differentiation of osteoclasts and chondrocytes in the mechanism [56]. However, meta-analysis of randomized controlled trials either in adults or in children showed no beneficial effects of vitamin D supplementation on bone health [57,58]. Red meat is an excellent source of protein. However, the actual influence of animal protein intake on bone may be moderated by dietary calcium intake and kidney function [42]. Data from the Korean National Health and Nutrition Examination Survey suggested that diets rich in milk, cereal and whole grain can protect against low bone mineral status in Korean male youths [59]. A prospective study including 599 mother–child pairs similarly found that a dietary pattern of infants aged 6 and 12 months characterized by a high intake of vegetables, fruit, meat/fish and other home-prepared foods had no association with the whole body bone mass of the infants at age four [60]. Considering the mutual and synergistic actions of foods and nutrients, the associations seen for similar dietary patterns across different studies may be inconsistent due to the inclusion of different food groups, different methods of bone measurements and other social or environmental factors.

The strength of this study was that we used a dietary pattern-based approach to assess the relationship between dietary factors and bone mineral status. This approach overcomes the limitations of traditional approaches that examine a single nutrient or food, as real diets are complex and include many different nutrients. There are also some potential limitations of the study. First, it is a cross-sectional design, and therefore causal relationships cannot be made. Second, as the participants were recruited by volunteers, our sample is not representative of all school-age children. Third, the sample size was relatively small, including 465 children (200 females and 265 males), and lacks sufficient power to detect associations with statistical significance. Finally, there may exist some residual confounding variables that were unmeasured or incorrectly measured in this study. Therefore, the results should be interpreted with caution, and a larger group of participants should be examined to confirm the findings.

## 5. Conclusions

Our findings suggested that diets rich in fruit, milk and eggs exert a beneficial effect on bone health in school-age children. Moreover, diets rich in animal organs and refined cereals are associated with bone loss, while a U-shaped association was seen between diets rich in beverages, fish and low-fat milk and bone mineral status. A healthy nutrient-balanced dietary pattern in childhood not only promotes peak BMC in puberty but also reduces osteoporosis and fracture risk in adulthood. Our study recommends home or school meals should be rich in fruit, milk, eggs with a moderate amount of vegetables, coarse grains and meat to promote bone development in school-age children. Furthermore, prospective or intervention studies are needed to confirm the relationship between dietary patterns and bone mass in the pediatric population. This study provides initial evidence that contributes to the development of novel dietary strategies for improving school-age children’s bone growth and preventing osteoporosis in later life.

## Figures and Tables

**Table 1 nutrients-14-03752-t001:** Baseline characteristics in participants divided by dietary patterns *.

	Dietary patterns					*p*-Value
	Vegetables-Whole Grains-Red Meat (*n* = 86)	Fruit-Milk-Eggs (*n* = 117)	Seafood-Mushrooms-Nuts (*n* = 82)	Beverages-Fish-Low-Fat Milk (*n* = 68)	Animal Organs-Refined Cereals (*n* = 112)	
Sex (*n*, %)						<0.001
boys	49, 57.0	75, 64.1	29, 35.4	37, 54.4	75, 67.0	
girls	37, 43.0	42, 35.9	53, 64.6	31, 45.6	37, 33.0	
Age (year)	8.2 (7.2–8.8)	7.9 (7.3–8.7)	7.9 (7.2–8.8)	8.1 (7.4–8.8)	8.1 (7.5–8.8)	0.693
Height (cm)	128.3 (122.9–137.0)	128.2 (124.0–134.2)	127.6 (122.1–135.2)	127.3 (122.2–133.3)	129.5 (122.9–134.4)	0.554
Weight (kg)	25.7 (20.8–32.2)	25.2 (21.6–28.5)	25.0 (21.4–29.2)	24.8 (21.8–28.8)	24.6 (22.2–28.6)	0.999
BMI z-score	−0.43 (−1.20–0.70)	−0.28 (−1.22–0.28)	−0.31 (−1.18–0.62)	−0.27 (−1.20–0.73)	−0.67 (−1.33–0.49)	0.495
Energy intake (Kcal)	1360 (1096–1669)	1402 (1096–1648)	1322 (1115–1707)	1383 (1132–1800)	1343 (1179–1672)	0.901
Protein intake (g)	59.6 (48.3–78.7)	59.5 (49.2–74.1)	63.1 (49.7–79.9)	66.6 (48.2–87.8)	56.4 (49.4–76.7)	0.604
Calcium intake (mg)	470.1 (361.8–652.3)	540.1 (409.9–667.8	483.5 (375.7–590.7)	522.7 (376.9–637.6)	424.9 (312.8–529.6)	<0.001
Vitamin D intake (IU)	80.0 (51.1–110.2)	87.0 (57.6–117.8)	84.8 (54.3–110.0)	93.5 (62.0–142.6)	74.1 (50.8–97.2)	0.025
Supplement intake of calcium (*n*, %)	26, 30.2	57, 48.7	35, 42.7	25, 36.8	46, 41.1	0.107
Supplement intake of multivitamins (*n*, %)	15, 17.4	21, 17.9	12, 14.6	13, 19.1	18, 16.1	0.952
Physical activities (MET×h/day)	38.8 (37.2–41.9)	39.5 (37.3–42.2)	39.0 (37.2–42.0)	38.0 (36.5–39.9)	39.6 (37.1–42.8)	0.050
Passive smoking (*n*, %)	24, 27.9	26, 22.2	18, 22.0	23, 33.8	28, 25.0	0.413
Delivery method (*n*, %)						0.147
Delivery	36, 41.9	59, 50.4	41, 50.0	30, 44.1	66, 58.9	
Cesarean section	50, 58.1	58, 49.6	41, 50.0	38, 55.9	46, 41.1	
Household income (*n*, %)						0.034
2000–5000	4, 4.7	12, 10.3	4, 4.9	4, 5.9	4, 3.6	
5001–8000	7, 8.1	9, 7.7	7, 8.5	13, 19.1	19, 17.0	
8001–12,000	10, 11.6	20, 17.1	18, 22.0	4, 5.9	19, 17.0	
12,001–15,000	14, 16.3	20, 17.1	11, 13.4	10, 14.7	16, 14.3	
>15,000	38, 44.2	39, 33.3	31, 37.8	20, 29.4	29, 25.9	
Unknown	13, 15.1	17, 14.5	11, 13.4	17, 25.0	25, 22.3	
BMC (g)						
Total body	922 (798–1049)	918 (851–1033)	919 (823–1026)	915 (840–989)	916 (841–986)	0.910
Total body less head	576 (490–684)	576 (516–661)	568 (505–666)	572 (526–633)	577 (503–635)	0.940
BMD (g/cm^2^)						
Total body	0.781 (0.725–0.836)	0.782 (0.742–0.833)	0.780 (0.733–0.825)	0.775 (0.740–0.815)	0.769 (0.740–0.808)	0.608
Total body less head	0.604 (0.552–0.668)	0.609 (0.572–0.656)	0.604 (0.569–0.645)	0.608 (0.567–0.636)	0.597 (0.564–0.643)	0.845

* Participants were classified into five groups by dietary patterns. The dietary pattern with the maximum score was selected as the main pattern for each participant. Values are presented as medians (interquartile range) or percentages. Physical activities are evaluated by metabolic equivalent (MET) hours per day. Passive smoking was defined as exposure to two cigarettes or more than 5 min per day on average.

**Table 2 nutrients-14-03752-t002:** Factor loading matrix of dietary patterns by principal component analysis with varimax rotation (*n* = 465) *.

Food Groups	Dietary Patterns				
	Vegetables- Whole Grains- Red Meat	Fruit-Milk- Eggs	Seafood- Mushrooms- Nuts	Beverages- Fish-Low-Fat Milk	Animal Organs- Refined Cereals
Whole grains	**0.569**				
Refined cereals	0.298				**0.455**
Soybean			0.264		
Fatty milk		**0.614**			
Low-fat milk				**0.599**	
Dark vegetables	**0.752**	0.223			
Light-colored vegetables	**0.757**				
Preserved vegetables					**0.401**
Light fruit	0.211	**0.662**	0.279		
Dark fruit		**0.536**	0.371		
Red meat	**0.457**	0.286			**0.417**
Animal organs			0.338		**0.564**
Poultry				0.213	0.367
Freshwater fish	0.218		0.328	**0.417**	
Sea fish	0.223			**0.673**	
Marinated animal food					0.360
Mollusks and shellfish			**0.605**		0.296
Eggs		**0.515**			
Mushrooms			**0.611**		
Nut			**0.445**		−0.204
Soup and beverage		0.222		**0.611**	0.327

* Factor loadings of <|0.20| are not listed in the table for simplicity. Loadings ≥|0.40| are in bold.

**Table 3 nutrients-14-03752-t003:** Multiple linear regression analysis between dietary patterns and bone mass.

Bone Measures	Dietary Patterns	Model 1		Model 2	
		*β*	95% CI	*β*	95% CI
TB BMC (g)	Vegetables-whole grains-red meat	14.180 *	(2.910, 25.449)	2.393	(−5.038, 9.824)
	Fruit-milk-eggs	19.329 **	(7.871, 30.788)	10.480 *	(2.190, 18.770)
	Seafood-mushrooms-nuts	2.924	(−7.386, 13.234	1.304	(−5.590, 8.197)
	Beverages-fish-low-fat milk	0.900	(−9.549, 11.350)	0.242	(−7.064, 7.548)
	Animal organs-refined cereals	−27.556 ***	(−38.817, −16.295)	−10.305 *	(−18.433, −2.176)
TBLH BMC (g)	Vegetables-whole grains-red meat	12.250 **	(3.267, 21.233)	1.776	(−3.020, 6.572)
	Fruit-milk-eggs	11.882 *	(2.694, 21.070)	5.577 *	(0.214, 10.941)
	Seafood-mushrooms-nuts	0.703	(−7.527, 8.932)	0.093	(−4.357, 4.543)
	Beverages-fish-low-fat milk	0.055	(−8.283, 8.394)	−0.490	(−5.206, 4.226)
	Animal organs-refined cereals	−21.366 ***	(−30.364, −12.367)	−6.346 *	(−11.596, −1.096)
TB BMD (g/cm^2^)	Vegetables-whole grains-red meat	0.008 **	(0.002, 0.013)	0.003	(−0.001, 0.008)
	Fruit-milk-eggs	0.009 **	(0.004, 0.015)	0.005	(0.000, 0.010)
	Seafood-mushrooms-nuts	0.003	(−0.002, 0.008)	0.001	(−0.003, 0.005)
	Beverages-fish-low-fat milk	0.001	(−0.004, 0.006)	−0.001	(−0.005, 0.003)
	Animal organs-refined cereals	−0.013 ***	(−0.019, −0.008)	−0.006 *	(−0.011, −0.001)
TBLH BMD (g/cm^2^)	Vegetables-whole grains-red meat	0.008 **	(0.003, 0.014)	0.002	(−0.001, 0.005)
	Fruit-milk-eggs	0.006 *	(0.001, 0.012)	0.002	(−0.001, 0.005)
	Seafood-mushrooms-nuts	0.001	(−0.004, 0.005)	0.000	(−0.003, 0.003)
	Beverages-fish-low-fat milk	0.000	(−0.005, 0.005)	−0.001	(−0.004, 0.002)
	Animal organs-refined cereals	−0.013 ***	(−0.018, −0.008)	−0.004 *	(−0.007, −0.001)

BMC: bone mineral content, BMD: bone mineral density, TB: total body, TBLH: total body less head. * *p* < 0.05, ** *p* < 0.01, *** *p* < 0.001. Model 1: adjusted for age, sex and energy intake. Model 2: further adjusted for height, weight, physical activity, household income, parental education, passive smoking, protein intake, calcium intake, vitamin D intake, supplement intake of calcium and supplement intake of multivitamins.

**Table 4 nutrients-14-03752-t004:** Covariate-adjusted mean of BMC and BMD by tertiles (T) of energy-adjusted dietary pattern scores *.

	T1 Mean ± SEM	T2 Mean ± SEM	T3 Mean ± SEM	%Diff	*p*-Diff	*p*-Trend
Vegetables-whole grains-red meat pattern						
TB BMC (g)	936.7 ± 5.70	929.0 ± 5.68	934.7 ± 5.76	−0.212	0.621	0.809
TBLH BMC (g)	588.8 ± 3.71	587.4 ± 3.70	588.0 ± 3.75	−0.144	0.962	0.874
TB BMD (g/cm^2^)	0.782 ± 0.003	0.782 ± 0.003	0.784 ± 0.003	0.256	0.900	0.722
TBLH BMD (g/cm^2^)	0.609 ± 0.002	0.613 ± 0.002	0.610 ± 0.002	0.164	0.560	0.778
Fruit-milk-eggs pattern						
TB BMC (g)	920.7 ± 5.89	938.6 ± 5.64	940.8 ± 5.83	2.178	0.037	0.021
TBLH BMC (g)	581.8 ± 3.84	588.5 ± 3.68	593.9 ± 3.80	2.077	0.100	0.033
TB BMD (g/cm^2^)	0.777 ± 0.004	0.787 ± 0.003	0.785 ± 0.003	1.030	0.103	0.119
TBLH BMD (g/cm^2^)	0.608 ± 0.002	0.611 ± 0.002	0.612 ± 0.002	0.658	0.459	0.235
Seafood-mushrooms-nuts pattern						
TB BMC (g)	929.2 ± 5.81	936.4 ± 5.74	934.7 ± 5.71	0.592	0.665	0.505
TBLH BMC (g)	587.2 ± 3.78	589.6 ± 3.74	587.4 ± 3.72	0.025	0.888	0.978
TB BMD (g/cm^2^)	0.780 ± 0.003	0.784 ± 0.003	0.784 ± 0.003	0.513	0.736	0.525
TBLH BMD (g/cm^2^)	0.611 ± 0.002	0.611 ± 0.002	0.609 ± 0.002	−0.327	0.819	0.592
Beverages-fish-low-fat milk pattern						
TB BMC (g)	945.6 ± 5.67	920.7 ± 5.65 ^^^^	934.0 ± 5.67	−1.222	0.009	0.155
TBLH BMC (g)	595.9 ± 3.69	579.6 ± 3.68 ^^^^	588.8 ± 3.69	−1.188	0.009	0.181
TB BMD (g/cm^2^)	0.789 ± 0.003	0.778 ± 0.003	0.781 ± 0.003	−1.014	0.087	0.120
TBLH BMD (g/cm^2^)	0.615 ± 0.002	0.607 ± 0.002	0.610 ± 0.002	−0.813	0.060	0.123
Animal organs-refined cereals pattern						
TB BMC (g)	937.8 ± 5.94	933.9 ± 5.73	928.7 ± 5.96	−0.972	0.586	0.303
TBLH BMC (g)	591.5 ± 3.87	588.2 ± 3.73	584.62 ± 3.87	−1.160	0.492	0.234
TB BMD (g/cm^2^)	0.785 ± 0.004	0.785 ± 0.003	0.778 ± 0.004	−0.892	0.272	0.191
TBLH BMD (g/cm^2^)	0.613 ± 0.002	0.612 ± 0.002	0.607 ± 0.002	−0.979	0.211	0.114

* Covariates are adjusted for age, sex, height, weight, physical activity, household income, parental education, passive smoking, energy intake, protein intake, calcium intake, vitamin D intake, supplement intake of calcium and supplement intake of multivitamins. BMC: bone mineral content, BMD: bone mineral density, TB: total body, TBLH: total body less head. %Diff: percentage difference of BMC between T3 and T1 = (T3 − T1)/T1 × 100%. *p*-diff: *p*-value for between-group difference analyzed by ANCOVA. ^^^^
*p* < 0.01 compared with T1.

## Data Availability

Not applicable.

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
