# Peer review of "The Relationship between Dietary Pattern and Bone Mass in School-Age Children"

_nutrients, 2022, doi:10.3390/nu14183752_

Round 1
Reviewer 1 Report
1. The concluding sentence of the Abstract just repeats the results. What are the implications of these findings?
2. Methods – Model 1 includes energy intake and Model 2 (which should just be a list in addition to Model 1) again lists energy intake.
3. The discussion of eggs as promoting bone says eggs are a good source of vitamin D. They are not high in vitamin D however. One would find more vitamin D in organ meats and in seafood, and those foods are in patterns where a positive effect on bone was not seen. Rewrite the discussion of where vitamin D might be impacting bone.
4. You cite reference [39]as evidence of protein’s positive effect on bone. There is no proof calbindin synthesis specifically is impacted by dietary protein (which for for IGF this has been shown). Bone is made of protein so the effect of protein may be more general – for anabolism.
5. You cite reference 50 in the sentence: “ However, a high intake of protein may exert detrimental effects on bone [50]” yet reference 50 says the opposite, that optimal protein is positive for bone in older adults and one should have more not less. There are a few references that do find detrimental effects of protein – they are old but you might still cite them, such as Feskanich D, Willett WC, Stampfer MJ, Colditz GA. Protein consumption and bone fractures in women. Am J Epidemiol. 1996 Mar 1;143(5):472-9. doi: 10.1093/oxfordjournals.aje.a008767. PMID: 8610662.
6. The Conclusion repeats the results and does not acknowledge the types of foods children should eat and how a change in intake could be done (given cultural considerations –e.g. are school meals an option? )
7. The Reference list is good except for 39, 41, 43, 45 which should be updated.
Reviewer 2 Report
Well designed and well written article however there are several issues which require explanation.
Major comments:
1. Linear regression models were adjusted for sex (model1) and also for household income (model 2) however it is necessary to add information whether household income which is so much different among study group was related to the specific dietary pattern; Information on whether the monthly income of 2000yuans reflects poverty is missing.
2. Despite different dietary patterns, girls and boys had very similar calcium and vitamin D intake and TBLH BMD but differed significantly in relation to energy and protein intake. It would be of value to know which percentage of girls/boys were identified with „the fruit-milk-eggs” pattern which was showed as „the best” and the one with „animal-organs, refined cereals” showed as unfavourable. It would be of interest to know which one of the two dietary patterns was preferred by girls and which one was preferred by boys.
3. For such small children data on BMI is not a representative index- instead the percentiles or BMI- Z scores should be presented.
